# Performance Analysis of Addressing Mechanisms in Inter-Operable IoT Device with Low-Power Wake-Up Radio

**DOI:** 10.3390/s19235106

**Published:** 2019-11-21

**Authors:** Taewon Song, Taeyoon Kim

**Affiliations:** 1IoT Connectivity Standard Team, Chief Technology Officer, LG Electronics, 19, Yangjae-daero 11-gil, Seocho-gu, Seoul 06772, Korea; taewon.song@lge.com; 2Department of Smart-car, Soonchunhyang University, 22 Soonchunhyang-ro, Shinchang-myeon, Asan-si, Chungcheongnam-do 31538, Korea

**Keywords:** low-power wake-up radio, power saving, bloom filter, IEEE 802.11ba

## Abstract

Internet of Things (IoT) technology is rapidly expanding the use of its application, from individuals to industries. Owing to this, the number of IoT devices has been exponentially increasing. Considering the massive number of the devices, overall energy consumption is becoming more serious. From this point of view, attaching low-power wake-up radio (WUR) to the devices can be one of the candidate solutions to deal with this problem. With WUR, IoT devices can go to sleep until WUR receives a wake-up signal, which enables a significant reduction of its power consumption. Meanwhile, one concern for WUR operation is the addressing mechanism, since operational efficiency of the wake-up feature can significantly vary depending on the addressing mechanism. We therefore introduce addressing mechanisms for IoT devices equipped with WUR and analyze their performances, such as elapsed time to wake up, false positive probability and power/energy consumption, to provide appropriate addressing mechanisms over practical environments for IoT devices with WUR.

## 1. Introduction

Internet of things (IoT) has been a mainstream technology in telecommunications. Currently, IoT devices are utilized on diverse systems, from consumer appliances to industrial applications. To support these various applications, there have been lots of standards-related activities in the IoT industry through various alliances and consortiums and many of them are quite mature in their own realm and territory [1,2,3,4,5]. As a result, this leads to a lack of inter-operability between these IoT systems resulting in a new type of interoperable platform. For example, Hewlett Packard Enterprise started supporting an IoT platform which can operate based on inter-operability with the help of OneM2M [6], which are designed to be applied in many different industries and take account of diverse inputs and requirements from any sector. As shown in Figure 1, multiple devices can be connected to diverse applications at the same time through the interoperable IoT platform. From this point of view, various IoT applications can be used as long as the minimum conditions are met, regardless of the hardware of the device.

On the one hand, raising this kind of platform inevitably causes a fundamental problem; a power consumption issue. On this platform, since large amounts of transmission from lots of applications to the device can occur, this can cause rapid battery exhaustion. This can especially be harmful for wireless devices with small battery capacity, such as mobile phones or smart watches.

To deal with the power consumption issue, several approaches to save their consumed power have been proposed. Two typical schemes among them are duty-cycle operation and low-power wake-up radio (WUR). In duty-cycle power saving operation, the radio can reduce its power consumption by periodically turning the transceiver on and off. Duty-cycle power saving operation has the advantage that no additional physical device is required; however, most of the duty-cycle protocols still suffer from overhearing and idle listening, which brings about redundant power consumption. For example, idle listening occurs when a station listens to wireless medium even when there is no communication and overhearing occurs when a station listens for communications destined for another node. These can be a critical aspect in energy-constrained wireless devices.

As an alternative to it, WUR has been proposed. In this wake-up radio system, an additional low-power radio is connected to the device along with conventional 802.11 radio. The purpose of WUR is to detect a wake-up signal sent by the wake-up transmitter before the wireless data communication destined for itself. In this way, the conventional radio can be configured to stay in the lowest power mode by default and can selectively listen to the wireless medium only when there is a signal to receive.

A representative standard to support wake-up radio is IEEE 802.11ba [7]. A task group of IEEE 802.11 was formed to develop draft 0.1 (D0.1) in 2017 and D4.0 has been released in 2019. Although the normative works of IEEE 802.11ba are just in the beginning stage, it can be applied to scenarios of smart homes/offices, warehouses, outdoor cattle farms, the synchronous wake-up of sensors, the synchronous wake-up of wearable devices, the reconnection of wearable devices, wake-up vehicle-to-pedestrian (V2P) radios and smart scanning.

Some aspects may be required to support the WUR. Since IEEE 802.11ba belongs to the 11 family, primary connected radio must support legacy IEEE 802.11. Additionally, the hardware needs to be newly deployed in order to be capable of the additional radio. However, since they are the world’s most widely used wireless computer networking standards, the main framework of the IoT platform does not need to completely be redesigned. Although it also requires an additional radio, overhearing and idle listening problems can be drastically alleviated. Due to the advantages compared to duty-cycle operation, this paper considers WUR a key solution for solving the power consumption problem of IoT devices and focuses on the detailed performance of WUR.

One of the things to consider when applying WUR is ’addressing’. Assuming a transmitter wants to wake up multiple devices, the transmitter may basically transmit a wake-up frame to recipients separately one by one using unicasting, which causes a significant delay in waking up the receivers. Otherwise, the transmitter can wake up all receivers at once using broadcast by risking redundant power consumption. Groupcasting can also be one of the solutions, but it requires a group assigning procedure in advance. The receivers that consistently wake up may be bonded together as a group, but it may be useless if the recipients are selected randomly.

Based on these addressing mechanisms, detailed addressing mechanisms, which can be adopted for various environments, have been studied. In Reference [8], unicasting, groupcasting and broadcasting wake-up mechanisms are realized by a wake-up signal containing the address and the mask. If all mask bits are set to 1, only one receiver with the same address will wake up. For broadcast, all bits in the mask are set to 0. That is, all nodes that receive this signal will wake up. To send a multicast wake up signal, a subset of the relevant bits is set to 1, which only wakes up the nodes in the subgroup. An addressing mechanism is proposed in Reference [9] that uses the duration of the signal falling-edge instead of decoding the full address of the wake-up signal. Specifically, each digit of the address can be represented by the duration between the falling-edges. In Reference [10], the authors proposed a bandpass filter bank which can combine the frequency domain and time domain addressing spaces to allow for the selective addressing of nodes.

By changing the way of thinking, a newly developed data structure for addressing is also worth considering. A Bloom filter is a space-efficient data structure, which can be used to test whether or not a certain element is included in a set. One of the most important features of the Bloom filter is that a false negative case does not occur. This means that there is no case where the station to be woken up does not wake up. Because of these characteristics, the Bloom filter has been utilized for various applications, such as content delivery network, blocking malicious URLs and wallet synchronization for cryptocurrencies. In Reference [11], the authors proposed a new time-dependent profile matching algorithm based on Bloom filters. Using this, a certain user can find another user that shares his/her interest in the proximity. As such, a Bloom filter can be a candidate mechanism for addressing. In References [12,13,14], the authors described the design and implementation of a wireless wake-up module that uses this addressing mechanism utilizing a Bloom filter.

In this work, we mathematically analyze addressing mechanisms based on IEEE 802.11ba, which is the most cutting-edge protocol for wake-up radio. Expected wake-up time, false positive probability, average throughput and expected power/energy consumption for the mentioned addressing mechanisms are analyzed. We perform extensive numerical analysis and present here our discussion on how to properly use the addressing mechanism in various environments.

The remainder of the paper is structured as follows. Section 2 introduces the operation of IEEE 802.11ba, which is the basis of the addressing mechanism to be analyzed and a theoretical background of Bloom filter. Next, in Section 3, we describe each addressing mechanism and analyze its delay until the recipients are woken up, its false wake-up probability and its mean power consumption. The listed addressing mechanisms are compared in Section 4. Finally, Section 5 concludes the paper.

## 2. Background

This section first introduces IEEE 802.11ba, a representative low-power wake-up protocol and its brief operational procedure. We next review a mathematical foundation for the standard Bloom filter for better understanding.

### 2.1. IEEE 802.11ba Operation

IEEE 802.11ba assumes an additional low-power wake-up radio, that is WUR, which aims only to wake up the conventional 802.11 radio. Hence, WUR only needs to receive wake-up frames and forward them to its conventional radio, resulting in very low power consumption. Furthermore, since IEEE 802.11ba is on 11-family, it can be easily adopted to existing wireless local area networks (WLANs).

Figure 2 shows the IEEE 802.11ba wake-up operation. In the existing Wi-Fi standard, a station just goes to sleep periodically to save its power while the station in the IEEE 802.11ba can be in a doze state unless there is a frame to receive since WUR can observe the channel. Once the MAC service data unit (MSDU) is delivered to an AP from the upper layer or a critical update or indication is needed for stations, the AP first transmits a wake-up frame. The wake-up frame is received by WUR of the station and the WUR prepares conventional 802.11 radio for a frame reception by waking it up. After that, the conventional 802.11 radio of the station may let the AP know that it is ready to receive data by sending a response frame (e.g., PS-Poll frame or Trigger-based frame) to the AP and this action can be a procedure of a conventional power saving mode. Consequently, after obtaining the opportunity to access the channel, a data transmission and subsequent acknowledgement can successfully be done. This channel access mechanism is a part of the conventional procedure of IEEE 802.11 and it can be either contention based or contention free.

Figure 3 shows a brief architecture of IEEE 802.11ba. On the transmitter side, a conventional 802.11 radio can transmit and receive the legacy IEEE 802.11 frames. Since the WUR does not support transmission, the transmitter does not need to be capable of reception of IEEE 802.11ba frames. On the receiver side, an additional radio is attached to the primary radio. The role of WUR is only to decode the wake-up frame and to wake up the primary transceiver. This transceiver for the receiver can be any kind of existing IEEE 802.11 standard device.

Since the WUR for IEEE 802.11ba is designed to have extremely low power consumption, it has to support a fairly low transmission rate. As a result, the wake-up frame is designed to be transmitted only at 62.5 kbps and 250 kbps. That is why the wake-up frame does not adopt the traditional IEEE 802.11 MAC protocol data unit (MPDU) format with 48-bit address fields. Instead, the Address field supports 12 bits and thus an address named WUR ID (WID) only for the wake-up procedure is newly defined. For the wake-up frames of unicast type, the Address field carries the WID assigned for the one WUR station by the AP to identify one WUR station. For the type of the multicast wake-up frame, the Address field carries the group ID (GID) assigned by the AP to identify one or more WUR stations. For broadcast wake-up frames and types of WUR beacons, the address field carries a transmitter ID (TXID) as it is a frame for all WUR stations. The newly defined IEEE 802.11ba wake-up frame format is shown in Figure 4.

### 2.2. A Review of Standard Bloom Filter

A Bloom filter is a data structure used to test whether an element is a member of a set. This property can be used in this paper to determine whether the address of a station is included. For the wake-up operation, the AP can indicate the target wake-up stations to the stations using the Bloom filter.

An empty Bloom filter can be represented as a bit array of *m* bits, which initially sets to 0. *k* independent hash functions are needed, each of which maps an address to one of the *m* array locations. Hence, to add an element into the Bloom filter, hash it to each of the *k* hash functions and thus *k* or less array positions can be drawn. Then the bits at all these positions are set to 1. Naturally, these hash functions must be shared between the AP and the stations associated to it to query its address.

Once the receiver gets the bit array, it will check whether its hashed address is included in the array or not. In order to test whether its address is in the set, the receiver performs the same procedure as the AP. It feeds it to each of the *k* hash functions to get *k* indexes, that is, the *k* hashed values of its address. If all bits are set to 1 for all *k* bit positions, then the receiver considers itself as an included one.

Since a Bloom filter is a probabilistic data structure, there is a chance that false positives occur, which means that stations that should not be indicated can be indicated by the filter. In particular, in cases when all bits derived from the *k* hash function are 1, then either the element is in the set, or the bits have by chance been set to 1 during the insertion of other elements, resulting in a false positive. This can incur redundant power consumption on the other hand, since a false negative never occurs in a Bloom filter by its inherited characteristics.

The above-mentioned simple procedures for initializing, inserting and querying a Bloom filter are shown in Figure 5. In this figure, the length of the Bloom filter, *m* is 10 and the number of hash functions, *k* is 3. All bits are set to 0 in the initialize phase. Each bit is numbered from 1 to 10, so a universe of all hash functions also distributes from 1 to 10. We assume Elements 1 and 2 are inserted into the filter and their hashed values are {2,4,6} and {1,8,9}, respectively. Logical ‘OR’ bit-wise operation is performed and thus the resulting Bloom filter becomes {1,1,0,1,0,1,0,1,1,0}, indicated as ‘insertion’ in Figure 5. Next, a recipient can query the existence of an element based on the Bloom filter. In case of querying Element 1, it calculates hashed values for the Element 1, which are 2,4,6 and check the Bloom filter at each index of the hashed value. Since all positions of the Bloom filter at {2,4,6} are 1s, the recipient can regard that the Element 1 is inserted in the Bloom filter. Through the same process, the recipient also thinks that Element 3, assuming whose hashed values are {2,4,10}, is not included in the Bloom filter and it is true as well. This case is called *true negative*. On the other hands, although Element 4 is not included in the Bloom filter in fact, the recipient will assume that Element 4 is included since the bits of Bloom filter at hashed values of Element 4, that is {1,2,6}, are all 1s. This can be called a *false positive* and it can cause redundant power consumption.

The false positive probability consists of the length of filter *m*, the number of hash functions *k* and the number of inserted elements *n*. Assuming hashed values are uniformly distributed among the space of the filter, the probability that a particular bit is not set to 1 by a particular hash function while inserting an element is $1-\frac{1}{m}$. If the hash functions have no correlation between each other, the probability that a particular bit is not set to 1 by the hash functions is ${\left(1-\frac{1}{m}\right)}^{k}$. Then if *n* elements are inserted, the probability that a particular bit is set to 1 is $1-{\left(1-\frac{1}{m}\right)}^{kn}$. This means that each of the *k* array positions computed by the hash function is 1 with above probability. Hence, the probability that the algorithm would incorrectly claim that the element is in the set, even the element is not intended to be inserted, that is, false positive probability, can be represented as
(1)$$p={\left(1-{\left[1-\frac{1}{m}\right]}^{kn}\right)}^{k}\approx {\left(1-e^{-kn/m}\right)}^{k}.$$

Based on Equation (Equation 1), assuming *m* and *n* are given, the number of optimal hash functions *k* that minimize the false positive probability can be derived as
(2)$$k=\frac{m}{n}\ln2.$$

In conclusion, at first the target false positive probability, tentative *p* has to be set. Next, in cases when the intended number of inserted elements is *n*, then the required number of hash functions, *k* can be calculated as in Equation (Equation 2). In sequence, if the target false positive probability *p* are given, then the optimal Bloom filter length can be calculated with Equation (Equation 1) as follows: (3)$$m=-\frac{n\ln p}{{\left(\ln2\right)}^{2}}.$$

## 3. Performance Analysis for Addressing Mechanisms

We compare the following four addressing mechanisms—(1) unicasting, (2) broadcasting, (3) groupcasting and (4) bloom-filter addressing. The beginning of this section first describes how each addressing mechanism can be designed based on the IEEE 802.11ba wake-up frame format. Next, we analyze some aspects of the mechanisms—wake-up delay, false positive probability and mean power/energy consumption. With wake-up delay, we can present which addressing mechanism is more suitable for delay-sensitive applications. We can also find the proper addressing mechanism for those applications that require a high data rate with an achievable data rate. False wake-up probability and mean power/energy consumption let us know the lifetime for the stations.

### 3.1. Design of Addressing Mechanisms

In the case of unicasting, two addressing methods are available. One is to use the Address field, which is mandatory for all APs and stations, and the other is to use Frame Body field, which can be optionally implemented. In the former case, the Address field of the wake-up frame contains a WID, which is negotiated between the AP and the station in advance, which identifies the station. In the latter case, the wake-up frame has a list of identifiers in the Frame Body field where one of the identifiers identifies the station. Since the maximum length for the Frame Body field is defined as 128 bits and the length for one WID is 12 bits, one wake-up frame can contain up to 10 individual addresses.

For broadcasting, an AP can wake up all stations using transmit ID, which is the address of the AP itself, which means that the wake-up frame is broadcasted, with a broadcasting wake-up frame. The AP may transmit this broadcasting wake-up frame to stations to indicate that a crucial update to the conventional 802.11 radio’s parameters is available via the radio for the stations.

A group ID is used for groupcasting. A group ID identifies a group of one or more stations. A wake-up frame with this group ID in the Address field is a group addressed wake-up frame that is addressed to all the WUR stations identified by that group ID. Although this groupcasting can be an efficient way for addressing, allocating group ID should be negotiated in advance and is actually defined as an optional feature in IEEE 802.11ba.

To use a Bloom filter, an appropriate length of the filter needs to be calculated. Once the AP determines how many stations it wakes up, denoted as *n*, and the false positive probability $p_{false}$ is given, the appropriate length can be calculated as seen in Equation (Equation 3). Meanwhile, using the Frame Body field, the maximum length of Bloom filter can be 128 bits. Hence, the AP may know how many wake-up frames to transmit, which can be represented as the calculated size of the Bloom filter, which is represented in Equation (Equation 3) divided by one Frame Body, $\left\lfloor m/128\right\rfloor$. In addition, the list of the used hash functions needs to be shared between the AP and the stations. Each hash function should also uniformly distribute inputs over a space of Bloom filter.

For unicasting wake-up, every station is woken up one by one. When the station’s WUR receives a wake-up frame assigned to it, it wakes up the conventional 802.11 radio in preparation for receiving a data frame. In the case of a broadcasting wake-up case, since the wake-up frame is received by all stations, the unscheduled station also wakes up the conventional 802.11 radio just like the scheduled station. In the groupcasting case, a station can also have group ID in addition to its own WID. If a station receives a wake-up frame which includes its WID or group ID, the station wakes up the conventional 802.11 radio. When the Bloom filter is used for the addressing method, a station should indicate how many hash functions are used. This information can be included in the wake-up frame. The station can query the existence of its address with this information. If the query proves that the address is in the filter, it wakes up the conventional 802.11 radio. Finally, it is assumed that the data frame and the (block)ack frame are transmitted to the recipients at the same time using MU-MIMO or OFDMA since these schemes are currently used in IEEE 802.11ax. These brief wake-up procedures for the addressing mechanisms described are illustrated in Figure 6.

### 3.2. Wake-Up Delay Analysis

In this subsection, wake-up delays are analyzed for each addressing method. We define the wake-up delay as the elapsed time from the first wake-up frame transmission to the completion of the ack frame transmission when the AP wants to wake up *n* stations. It also assumes that the wake-up frame follows the format of Figure 4 with a 128-bit frame body.

Referring to Figure 6, the average wake-up delay to wake up *n* stations, $E[T_{delay,uni}\left(n\right)]$ for the unicasting can be expressed as
(4)$$\begin{array}{c} E\left[T_{delay,uni}\left(n\right)\right]=n\cdot \left(T_{wakeupframe}+T_{sifs}\right)+E\left[T_{contention}\right]+E\left[T_{dataframe}\right]+T_{sifs}+T_{ackframe}, \end{array}$$
where $T_{wakeupframe}$, $T_{sifs}$, $T_{contention}$, $T_{dataframe}$ and $T_{ackframe}$ denote the duration for the transmission time of wake-up frame, the short inter-frame space (SIFS), contention, the transmission time of data frame and the transmission time of ack frame, respectively. $T_{wakeupframe}$, $T_{dataframe}$ and $T_{ackframe}$ can vary depending on their transmit rate and frame length. If channel acquisition is deterministically occurred or the length of data does not vary, $E[T_{contention}]$ and $E[T_{dataframe}]$ can be represented as $T_{contention}$ and $T_{dataframe}$, respectively.

In broadcasting, since all the stations will wake up at once, average wake-up delay, $E[T_{delay,broad}\left(n\right)]$ can be represented as
(5)$$E\left[T_{delay,broad}\left(n\right)\right]=T_{wakeupframe}+T_{sifs}+E\left[T_{contention}\right]+E\left[T_{dataframe}\right]+T_{sifs}+T_{ackframe}.$$

In the case of groupcasting, the average wake-up delay to wake up *n* stations, $E[T_{delay,group}\left(n\right)]$ can be represented as
(6)$$\begin{array}{ll} E[T_{delay,group}\left(n\right)]= & E\left[N_{wakeupframe,group}\left(n\right)\right]\cdot \left(T_{wakeupframe}+T_{sifs}\right)+\\ & E\left[T_{contention}\right]+E\left[T_{dataframe}\right]+T_{sifs}+T_{ackframe}, \end{array}$$
where $N_{wakeupframe,group}\left(n\right)$ denotes the number of wake-up frame, that is, the number of groups we need to call to wake up *n* targeting stations. We derive the expected value of $N_{wakeupframe,group}\left(n\right)$ because even if the *n* is the same among the cases, the number of groups may vary depending on which stations in which group are subject to wake-up. To derive $N_{wakeupframe,group}\left(n\right)$, we need to define some parameters and present the number of all stations as $N_{station}$ and the size of a group as $L_{group}$. For ease of analysis, the set of all stations is divided in order, which is the size of $L_{group}$.

So, the number of groups becomes $N_{group}=N_{station}/L_{group}$. For example, if there are 20 stations, numbered from 1 to 20 and the number of groups is 5, then the grouping is as follows: Group 1 = {1,2,3,4}, Group 2 = {5,6,7,8}, Group 3 = {9,10,11,12}, Group 4 = {13,14,15,16}, Group 5 = {17,18,19,20}. In this case, $N_{group}$, $N_{station}$, and $L_{group}$ become 5, 20, and 4, respectively.

We can express $N_{wakeupframe,group}\left(n\right)$ as a recurrence relation. Obviously, $N_{wakeupframe,group}\left(0\right)=0$ and $N_{wakeupframe,group}\left(1\right)=1$. If *n* becomes 2, $N_{wakeupframe,group}\left(2\right)$ can be either 1 or 2. If the second selected element belongs to the same group as the first selected element, $N_{wakeupframe,group}\left(2\right)=1$ and otherwise, $N_{wakeupframe,group}\left(2\right)=2$. Thus, in general, if the n+1th selected element belongs to a group that is already associated with the previous element, the $N_{wakeupframe,group}\left(n+1\right)$ is the same as $N_{wakeupframe,group}\left(n\right)$. Hence, $N_{wakeupframe,group}\left(n+1\right)$ can be expressed as follows:
(7)$$N_{wakeupframe,group}\left(n+1\right)=\left\{\begin{array}{ll} N_{wakeupframe,group}\left(n\right), & w.r.t.\;\frac{N_{wakeupframe,group}\left(n\right)\cdot L_{group}-n}{N_{station}-n}\\ N_{wakeupframe,group}\left(n\right)+1, & w.r.t.\;\frac{N_{station}-N_{wakeupframe,group}\left(n\right)\cdot L_{group}}{N_{station}-n}. \end{array}\right.$$

Thus, a recurrence relation between $E[N_{wakeupframe,group}\left(n+1\right)]$ and $E[N_{wakeupframe,group}\left(n\right)]$ can be expressed as
(8)$$\begin{array}{ll} E[N_{wakeupframe,group}\left(n+1\right)] & =E\left[N_{wakeupframe,group}\left(n\right)\right]+\frac{N_{station}-E\left[N_{wakeupframe,group}\left(n\right)\right]\cdot L_{group}}{N_{station}-n}\\ & =\left(1-\frac{L_{group}}{N_{station}-n}\right)\cdot E\left[N_{wakeupframe,group}\left(n\right)\right]+\frac{N_{station}}{N_{station}-n} \end{array}.$$

Let $1-\frac{L_{group}}{N_{station}-n}=f_{n}$ and $\frac{N_{station}}{N_{station}-n}=g_{n}$, then Equation (Equation 8) can be expressed as
(9)$$\begin{array}{c} E\left[N_{wakeupframe,group}\left(n+1\right)\right]-f_{n}\cdot E\left[N_{wakeupframe,group}\left(n\right)\right]=g_{n} \end{array}.$$

Multiplying both sides of Equation (Equation 9) by $\frac{1}{\prod_{k=0}^{n}f_{k}}$, then
(10)$$\begin{array}{c} \frac{E[N_{wakeupframe,group}\left(n+1\right)]}{\prod_{k=0}^{n}f_{k}}-\frac{f_{n}\cdot E\left[N_{wakeupframe,group}\left(n\right)\right]}{\prod_{k=0}^{n}f_{k}}=\frac{g_{n}}{\prod_{k=0}^{n}f_{k}} \end{array}.$$

Assuming $A_{n}=\frac{E[N_{wakeupframe,group}\left(n\right)]}{\prod_{k=0}^{n-1}f_{k}}$, then
(11)$$A_{n+1}-A_{n}=\frac{g_{n}}{\prod_{k=0}^{n}f_{k}}$$
and
(12)$$A_{0}=1.$$

With Equations (Equation 11) and (Equation 12), the general form of $A_{n}$ can be represented as
(13)$$A_{n}=1+\sum_{m=0}^{n-1}\frac{g_{m}}{\prod_{k=0}^{m}f_{k}}.$$

Hence, the general form of the expected number of wake-up frames to wake up *n* stations can be represented as follows: (14)$$\begin{array}{c} E\left[N_{wakeupframe,group}\left(n\right)\right]=\left(\prod_{k=0}^{n-1}f_{k}\right)\left(\sum_{m=0}^{n-1}1+\frac{g_{m}}{\prod_{k=0}^{m}f_{k}}\right). \end{array}$$

As a result, the average wake-up delay for groupcasting can be expressed with Equations (Equation 6) and (Equation 14).

In the case of the Bloom filter, the average wake-up delay to wake up *n* stations, $E[T_{delay,bf}\left(n\right)]$ can be represented as
(15)$$\begin{array}{ll} E[T_{delay,bf}\left(n\right)]= & N_{wakeupframe,bf}\left(n\right)\cdot \left(T_{wakeupframe}+T_{sifs}\right)+\\ & E\left[T_{contention}\right]+E\left[T_{dataframe}\right]+T_{sifs}+T_{ackframe}, \end{array}$$
where $N_{wakeupframe,bf}\left(n\right)$ denotes the number of wake-up frame to wake up *n* stations for the Bloom filter addressing. Similar to the case of groupcasting, $N_{wakeupframe,bf}\left(n\right)$ can vary depending on the target false positive probability. Generally, if the target false positive probability is smaller, the required length of the bloom filter becomes longer and vice versa.

As shown in Equation (Equation 3), the optimal length of Bloom filter can be calculated as follows: (16)$$m\left(n\right)=-\frac{n\ln p_{target}}{{\left(\ln2\right)}^{2}},$$
where m(n) is the optimal length of Bloom filter for waking up *n* stations and $p_{target}$ is the target false positive probability. As a result, the required number of wake-up frames can be calculated as(17)$$E\left[N_{wakeupframe,bf}\left(n\right)\right]=\left\lceil \frac{m(n)}{128}\right\rceil .$$

As a result, average wake-up delay for Bloom filter can be expressed with Equations (Equation 15) and (Equation 17).

### 3.3. False Positive Probability Analysis

In this subsection, false positive probabilities are analyzed for addressing mechanisms. Since the above-mentioned four addressing mechanisms do not suffer from false negative probability unless there is a transmission failure, only false positive probabilities are addressed in this section. In this paper, false positives are defined as the probability that a station wakes up even if it is not actually marked in received wake-up frames. This probability can particularly be important for the amount of power consumption.

In the case of unicasting, since wake-up frames are transmitted only to the corresponding stations, false positive probability
(18)$$p_{false,uni}\left(n\right)=0$$
for all *n*’s.

For broadcasts, the address field of a wake-up frame indicates that the station receiving the frame should wake up, so all stations wake up. In that sense, false positive probability
(19)$$p_{false,broad}\left(n\right)=1$$
for all *n*’s.

For groupcasting, the probability of false positives can vary depending on whether other stations in the group to which the station belongs should wake up. By the pigeonhole principle,
(20)$$p_{false,group}\left(n\right)=1,\;\mathrm{if}\;n>N_{station}-L_{group}.$$

On the other hand, if $n\leq N_{station}-L_{group}$, then $p_{false,group}\left(n\right)$ is the probability that all *n* target stations are not associated with the group that the particular station is associated with. For example, if the particular station belongs to Group 1, false positive does not occur only if the groups on the *n* target stations are not all of group 1. Hence, what is mentioned above can be represented as
(21)$$p_{false,group}\left(n\right)=1-p_{nofalse,group}\left(n\right),\;\mathrm{if}\;n\leq N_{station}-L_{group}.,$$
where $p_{nofalse,group}\left(n\right)$ is the probability that no false positive occurs. So Equation (Equation 21) becomes as follows: (22)$$\begin{array}{ll} p_{false,group}\left(n\right) & =1-p_{nofalse,group}\left(n\right)\\ & =1-\frac{N_{station}-L_{group}}{N_{station}-1}\cdot \frac{N_{station}-L_{group}-1}{N_{station}-2}\cdot \cdots \frac{N_{station}-L_{group}-\left(n-1\right))}{N_{station}-n}\\ & =1-\frac{\left(N_{station}-L_{group}\right)!\left(N_{station}-n-1\right)!}{\left(N_{station}-L_{group}-n\right)!\left(N_{station}-1\right)!}. \end{array}$$

For the Bloom filter, the difference between the other mechanisms is that the target false positive probability is set in advance, and then the actual false positive probability is calculated based on that probability. As defined earlier in Equation (Equation 16), let the target false positive probability $p_{target}$. Then we show that the total length of the Bloom filter is $m_{optimum}\left(n\right)=-\frac{n\ln p_{target}}{{\left(\ln2\right)}^{2}}$. It, however, is not the actual length of the Bloom filter but the optimal value of it, since we assume that the wake-up frame has fixed length, 128 bits, of the Frame Body field. Hence the actual length of the Bloom filter when *n* stations are indicated, $m_{actual}\left(n\right)$ can be calculated as
(23)$$m_{actual}\left(n\right)=128*\left\lceil -\frac{n\ln p_{target}}{128*{\left(\ln2\right)}^{2}}\right\rceil .$$

With Equations (Equation 1), (Equation 2) and (Equation 23), the actual false positive probability can be expressed as follows: (24)$$p_{false,bf}\left(n\right)={\left(1-e^{-\ln2}\right)}^{\frac{m_{actual}\left(n\right)\ln2}{n}}={\left(\frac{1}{2}\right)}^{\frac{m_{actual}\left(n\right)\ln2}{n}}.$$

### 3.4. Expected Energy/Power Consumption Analysis

In this subsection, the expected energies for one station per one cycle are analyzed. The expected power can be calculated by the expected energy divided by the expected wake-up delay. We evaluate these values considering the ‘true positive’ stations as well as the ‘false positive’ ones by using the equations calculated above.

In common, by the law of total expectation, the expected energy per one cycle when the number of wake-up stations is *n*, as shown in Figure 6, can be expressed as follows: (25)$$\begin{array}{c} E[energy(n)]\\ \;=E[energy(n)|wakeup]P(wakeup)+E[energy(n)|nowakeup]P(nowakeup)\\ \;=E\left[energy\left(n\right)|wakeup\right]\left\{P(wakeup|target)\cdot P(target)+P(wakeup|notarget)\cdot P(notarget)\right\}\\ \;+E\left[energy\left(n\right)|nowakeup\right]\left\{P(nowakeup|target)\cdot P(target)+P(nowakeup|notarget)\cdot P(notarget)\right\}\\ \;=E\left[energy\left(n\right)|wakeup\right]\left\{\frac{n}{N_{station}}+p_{false}\cdot \frac{N_{station}-n}{N_{station}}\right\}\\ \;+E\left[energy\left(n\right)|nowakeup\right]\left\{\left(1-p_{false}\right)\cdot \frac{N_{station}-n}{N_{station}}\right\}, \end{array}$$
where E[energy(n)] stands for the expected consumed energy when *n* stations are subject to wake up, wakeup is the event of wake-up and nowakeup is the event of no wake-up. target is the event that the AP decides to wake up a particular station and notarget is the event that the AP do not wake up the station. In detail, P(wakeup|target)=1 and P(nowakeup|target)=0 since there is no ‘false negative’ case for all addressing mechanisms. P(wakeup|notarget) is the false positive probabilities, $p_{false}$, as shown in Equations (Equation 18), (Equation 19), (Equation 22) and (Equation 24) according to the addressing mechanism. Since we assume that the target stations are selected randomly and uniformly, P(target) and P(notarget) become $\frac{n}{N_{station}}$ and $\frac{N_{station}-n}{N_{station}}$, respectively.

After that, the expected power for one station when the number of wake-up stations is *n* can be expressed as follows: (26)$$E\left[power\left(n\right)\right]=\frac{E[energy(n)]}{E[T_{delay}\left(n\right)]}$$
for each addressing mechanism.

Referring to Figure 2, in the case of unicasting, the expected consumed energy if the station is decided to wake up, $E[energy_{uni}\left(n\right)|wakeup]$, can be represented as follows: (27)$$\begin{array}{ll} E[energy_{uni}\left(n\right)|wakeup] & =\left\{\left(T_{wakeupframe}+T_{sifs}\right)P_{rx,wur}\right\}\\ & +\frac{n-1}{2}\left\{\left(T_{wakeupframe}+T_{sifs}\right)P_{rx,pcr}\right\}\\ & +\left(E[T_{contention}]+E[T_{dataframe}]+T_{sifs}\right)P_{rx,pcr}\\ & +\left(T_{ackframe}+T_{sifs}\right)P_{tx,pcr}, \end{array}$$
where $P_{rx,wur}$, $P_{rx,pcr}$, and $P_{tx,pcr}$ are the consumed power of reception for WUR, of reception for conventional 802.11 radio, and of transmission for conventional 802.11 radio, respectively. Consequently, the expected consumed energy decided not to wake, $E[energy_{uni}\left(n\right)|nowakeup]$ can be represented as follows: (28)$$\begin{array}{ll} E[energy_{uni}\left(n\right)|nowakeup] & =n\cdot \left(T_{wakeupframe}+T_{sifs}\right)P_{rx,wur}\\ & +\left(E\left[T_{contention}\right]+E\left[T_{dataframe}\right]+2T_{sifs}+T_{ackframe}\right)P_{idle,wur}, \end{array}$$
where $P_{idle,wur}$ is the expected power consumption when WUR, as well as conventional 802.11 radio, are in idle state. In conclusion, with Equations (Equation 25), (Equation 27), (Equation 28) and $p_{false,uni}=0$, as shown in Equation (Equation 18), the expected energy consumption can be calculated. With the calculated energy consumption and Equation (Equation 4), the expected power consumption can be calculated as well.

In the case of broadcasting, the expected consumed energy if the station is to be woken up, $E[energy_{broad}\left(n\right)|wakeup]$, can be represented as follows: (29)$$\begin{array}{ll} E[energy_{broad}\left(n\right)|wakeup] & =\left(T_{wakeupframe}+T_{sifs}\right)P_{rx,wur}\\ & +\left(E\left[T_{contention}\right]+E\left[T_{dataframe}\right]+T_{sifs}\right)P_{rx,pcr}\\ & +\left(T_{ackframe}+T_{sifs}\right)P_{tx,pcr}. \end{array}$$

Since all stations have to wake up in broadcasting, it is unnecessary to express $E[energy_{broad}\left(n\right)|nowakeup]$. Then, with Equations (Equation 25) and (Equation 29) with $p_{false,broad}=1$, the expected energy consumption in case of broadcasting can be calculated. With the calculated energy consumption and Equation (Equation 5), the expected power consumption can also be calculated.

For groupcasting, the expected consumed energy if the station is decided to wake up, $E[energy_{group}\left(n\right)|wakeup]$, can be represented as follows: (30)$$\begin{array}{ll} E[energy_{group}\left(n\right)|wakeup] & =\left\{\left(T_{wakeupframe}+T_{sifs}\right)P_{rx,wur}\right\}\\ & +\frac{E[N_{wakeupframe,group}\left(n\right)]-1}{2}\left\{\left(T_{wakeupframe}+T_{sifs}\right)P_{rx,pcr}\right\}\\ & +\left(E[T_{contention}]+E[T_{dataframe}]+T_{sifs}\right)P_{rx,pcr}\\ & +\left(T_{ackframe}+T_{sifs}\right)P_{tx,pcr}. \end{array}$$

Consequently, $E[energy_{group}\left(n\right)|nowakeup]$, can also be represented as follows: (31)$$\begin{array}{ll} E[energy_{group}\left(n\right)|nowakeup] & =E\left[N_{wakeupframe,group}\left(n\right)\right]\cdot \left(T_{wakeupframe}+T_{sifs}\right)P_{rx,wur}\\ & +\left(E\left[T_{contention}\right]+E\left[T_{dataframe}\right]+2T_{sifs}+T_{ackframe}\right)P_{idle,wur}. \end{array}$$

Hence, with Equations (Equation 25), (Equation 30) and (Equation 31) with $p_{false,group}\left(n\right)$, the expected consumed energy for groupcasting can be expressed. In the same way as above, the expected consuming power can also be obtained.

In the case of the Bloom filter, the expected consumed energy if the station is decided to wake up, $E[energy_{bf}\left(n\right)|wakeup]$, can be represented as follows: (32)$$\begin{array}{ll} E[energy_{bf}\left(n\right)|wakeup] & =\left\{\left(T_{wakeupframe}+T_{sifs}\right)P_{rx,wur}\right\}\\ & +\frac{E[N_{wakeupframe,bf}\left(n\right)]-1}{2}\left\{\left(T_{wakeupframe}+T_{sifs}\right)P_{rx,pcr}\right\}\\ & +\left(E[T_{contention}]+E[T_{dataframe}]+T_{sifs}\right)P_{rx,pcr}\\ & +\left(T_{ackframe}+T_{sifs}\right)P_{tx,pcr}. \end{array}$$

Consequently, $E[energy_{bf}\left(n\right)|nowakeup]$, can also be represented as follows: (33)$$\begin{array}{ll} E[energy_{bf}\left(n\right)|nowakeup] & =E\left[N_{wakeupframe,bf}\left(n\right)\right]\cdot \left(T_{wakeupframe}+T_{sifs}\right)P_{rx,wur}\\ & +\left(E\left[T_{contention}\right]+E\left[T_{dataframe}\right]+2T_{sifs}+T_{ackframe}\right)P_{idle,wur}. \end{array}$$

Hence, with Equations (Equation 17), (Equation 25), (Equation 32) and (Equation 33) with $p_{false,bf}\left(n\right)$, we can get the expected consumed energy as well as the expected consuming power for the Bloom filter.

## 4. Simulation Results

In this simulation, we omit channel contention between wake-up frames and data frame since the channel contention does not affect the result of the wake-up procedure. Hence, the elapsed time to contend for the channel is excepted from this simulation. We assume the length of the data frame and ack frame from conventional 802.11 radio to be 3000 Bytes and 32 Bytes, respectively. We also assume the length of Frame Body field for the Bloom filter to be 128 bits. Data rates of conventional 802.11 radio and WUR are 8.6 Mbps and 250 kbps, respectively. Transmit and receive power consumption [15], time parameters and groupcasting parameters for each radio and each state are shown in Table 1. To implement the Bloom filter flexibly to the number of elements, an arbitrary number of optimal hash functions should be shared by the inserter and the recipient. We assume universal hashing [16], selecting a hash function at random from a family of hash functions. According to the universal hashing, once an AP and stations share an array of numbers or vectors, the station can query the address with only the number of hash functions used. Other parameters not mentioned follow the IEEE 802.11ax standard, which is the latest standard for WLANs.

### 4.1. Expected Wake-Up Time and Average Data Throughput

Hereafter, the target stations mean that the station is subject to wake up by wake-up frame. Figure 7a shows the expected delays for the above-mentioned addressing mechanisms when the number of target stations varies. We define the expected delay as the average elapsed time from when the first wake-up frame is sent until the ack frame is sent.

It is straightforward that the expected delay for unicasting linearly increases. This is because, as the number of target stations increases by one, the elapsed wake-up time also increases by $T_{wakeupframe}$ plus $T_{sifs}$. In contrast, the expected delay for broadcasting is a constant value. This is because the wake-up procedure for broadcasting does not change regardless of how many target stations wake up. In the case of groupcasting and Bloom filter, the elapsed wake-up time gradually increases, but the slope of the groupcasting is steeper than the slope of the bloom filter.

With the elapsed time and the frame length, we assumed 3000 Bytes, the average data throughputs can also be analyzed. Figure 7b shows the average throughputs for the addressing mechanisms. Broadcasting maintains its data throughputs over the number of target stations. This is because only one wake-up frame is always transmitted for all situations. It can be seen that throughputs of all addressing mechanisms, except broadcasting, gradually decrease as the number of target stations increase. In the case of the Bloom filter, the throughput decreases in a cascading manner. It is because the number of required wake-up frames increases by one if the calculated required length of the Bloom filter becomes a multiple of the length of the Frame Body field, which is defined to be 128 bits. In the case of unicasting and groupcasting, data throughput gradually decreases because the number of target wake-up frames increases as the number of target stations increases.

### 4.2. False Positive Probability

Figure 8 shows false positive probabilities when the number of target stations varies. Since it is obvious that the false positive probabilities for unicasting and broadcasting are 0 and 1 at all cases, respectively, these are omitted in this figure.

As the number of target stations exceeds 100, the false positive probability for groupcasting becomes nearly 1 since almost all groups are indicated. As mentioned earlier, even if only one station is subject to be woken up, all stations in the group to which it belongs must wake up. This is why the curve of groupcasting converges on 1 as the number of target stations increases. For Bloom filters, the probability of false positives increases until the number of target stations reaches 80. It can be seen that the probability drastically drops when the number of target stations is 100 because the number of wake-up frames increases by 1, which lengthens the length of the Bloom filter. As shown in Equation (Equation 24), the false positive probability $p_{false,bf}\left(n\right)$ will decrease as the actual length of the Bloom filter $m_{actual}\left(n\right)$ increases. This situation also occurs when the number of target stations is around 160.

### 4.3. Mean Power and Energy Consumption

Power consumption is one of the good criteria to decide an appropriate addressing mechanism especially for the power-insufficient devices, such as mobile phones or small sensors. Figure 9a shows mean power consumption for addressing mechanisms. This mean power consumption stands for the consumed energy per unit time for one station. While the mean power for unicasting, groupcasting and broadcasting stays at a similar value over the variations of the target number, the mean power of the Bloom filter tends to increase as the target number increases. This is because the number of wake-up frames does not increase even if the number of target stations increases from 10 to 80 and it brings about the rise in false positive probability. Hence, some unintended stations become woken up and it causes redundant power consumption without lengthened wake-up delay. On the other hand, as the number of targets increases, so does the number of wake-up frames; the mean consuming power does not increase.

In Figure 9b, the consumed energy per one cycle for one station is shown. This one cycle stands for the time from when the first wake-up frame is sent until the ack frame is sent. Since the wake-up delay becomes longer and longer as the target number increases for unicasting and groupcasting, the energy consumption per one cycle is bound to increase. On the contrary, the energy consumption for broadcasting and the Bloom filter remains at a similar value over the target number since the number of wake-up frames will remain at 1 in the broadcasting case or increases to a very small number in the Bloom filter case.

From these analyses, unicasting wake-up is relatively inferior to other addressing mechanisms because unicasting not only takes longer to wake up, but also consumes more energy than other mechanisms. The unicasting wake-up can be a good option when the number of target stations is very small. Broadcasting wake-up is appropriate for stations that use applications requiring high data rates but have sufficient battery power, especially when there are lots of target stations. Groupcasting can be one of the addressing options, however, its false positive probability gets high when the target number is large. It can be solved by delicate grouping in advance; however, if the wake-up stations are randomly chosen, it becomes hard to adopt groupcasting. Using the Bloom filter, the expected wake-up delay is quite short while data throughput is relatively high. Hence, the Bloom filter can be a good addressing mechanism under ordinary environments.

### 4.4. Effect of Target False Positive Probability

In Figure 10a–c, we compare expected wake-up delay, actual false positive probability and mean power consumption when the target false positive probability, $p_{target}$ varies for the Bloom filter. If $p_{target}$ increases, the required length of the Bloom filter decreases and thus the expected wake-up delay becomes short. On the other hand, with the increased actual false positive probability, mean power consumption also becomes higher compared to low $p_{target}$ cases. With these results, $p_{target}$ value can be set depending on what characteristics the application has. Moreover, since $p_{target}$ value does not need to be negotiated between the AP and the stations, the AP can adjust the value on demand.

## 5. Conclusions

In this paper, we give intensive mathematical analysis among four addressing mechanisms which can be used for IoT devices with low-power wake-up radio are analyzed. Through the performance analyses, unicasting wake-up is a relatively inferior to other addressing mechanisms because unicasting not only takes longer to wake up but also consumes more energy than other mechanisms. The unicasting wake-up can be a good option when the number of target stations is very small. Broadcasting wake-up is appropriate for stations that use applications requiring high data rates but have sufficient battery power, especially when there are lots of target stations. Groupcasting can be one of the addressing options, however, its false positive probability is high when the target number is large. It can be solved by delicate grouping in advance; however, if the wake-up stations are randomly chosen, it becomes hard to adopt groupcasting. Using the Bloom filter, the expected wake-up delay is quite short while data throughput is relatively high. Further, by adjusting the target false positive probability, Bloom filters can be fine tuned for specific traffic and application conditions. Hence, the Bloom filter can be a good addressing mechanism under ordinary environments. According to our study, an appropriate addressing mechanism can be selected depending on the application characteristics.

## Figures and Tables

**Figure 1 sensors-19-05106-f001:**
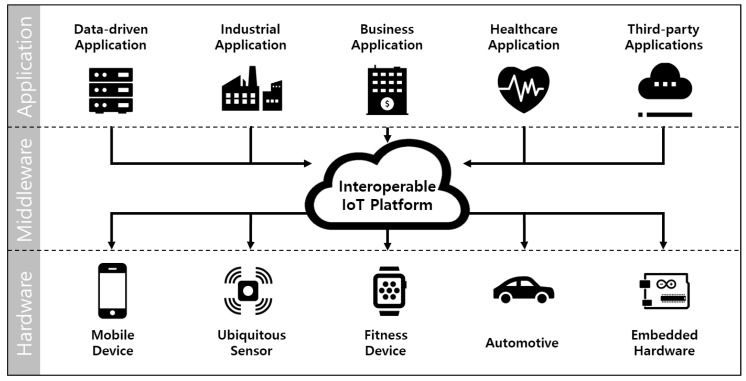
Simple topology for interoperable IoT platform.

**Figure 2 sensors-19-05106-f002:**
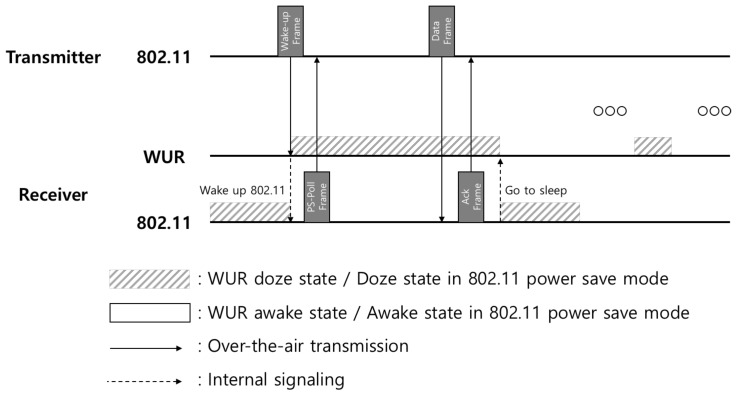
Timing diagram for IEEE 802.11ba operation.

**Figure 3 sensors-19-05106-f003:**
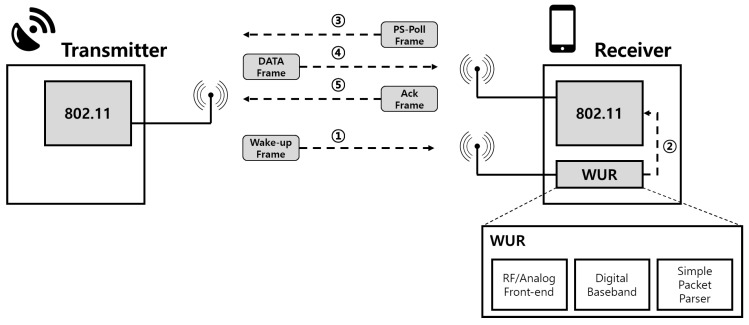
Architecture of IEEE 802.11ba.

**Figure 4 sensors-19-05106-f004:**
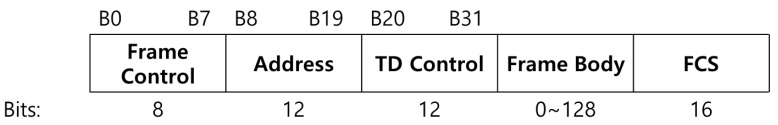
IEEE 802.11ba wake-up frame structure.

**Figure 5 sensors-19-05106-f005:**
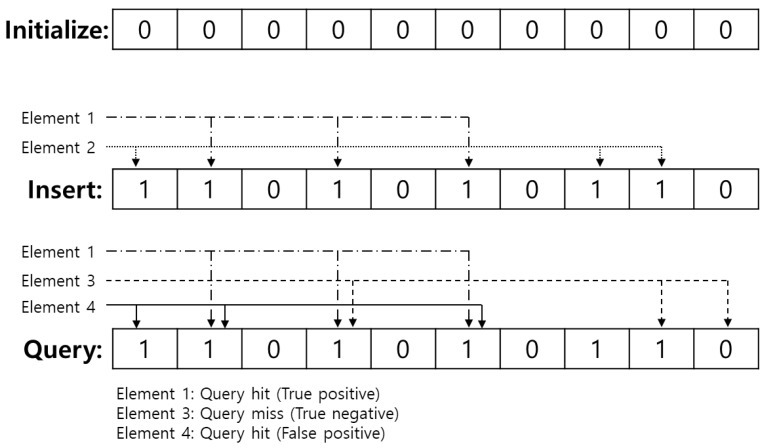
Initialize, Insertion and Query for Bloom Filter.

**Figure 6 sensors-19-05106-f006:**
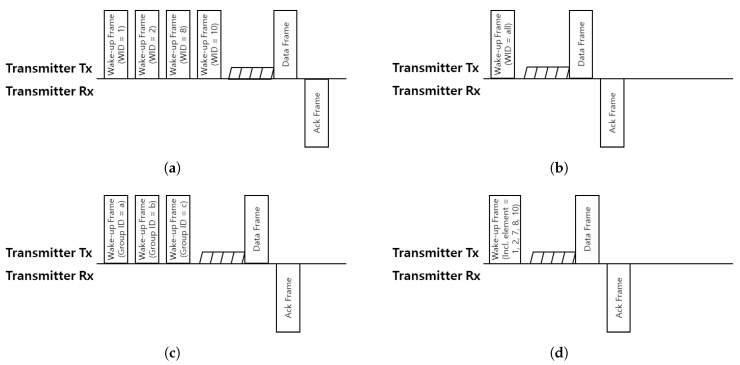
Wake up and data transmission procedure with wake up radio (WUR) using (**a**) unicasting, (**b**) broadcasting, (**c**) groupcasting and (**d**) Bloom filter. The transmitter wakes up recipients based oneach addressing method and transmits data frame to them.

**Figure 7 sensors-19-05106-f007:**
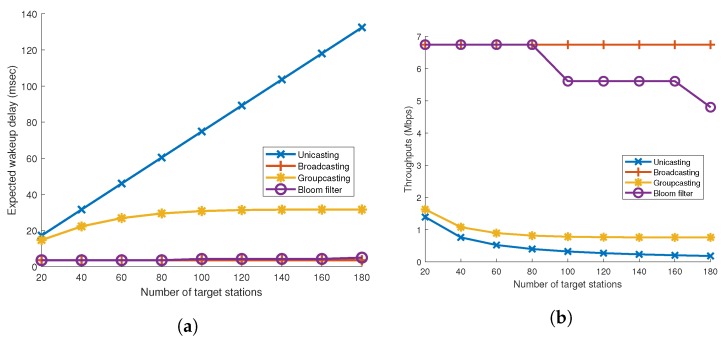
(**a**) Expected wake-up delay and (**b**) average data throughput according to the number of target stations.

**Figure 8 sensors-19-05106-f008:**
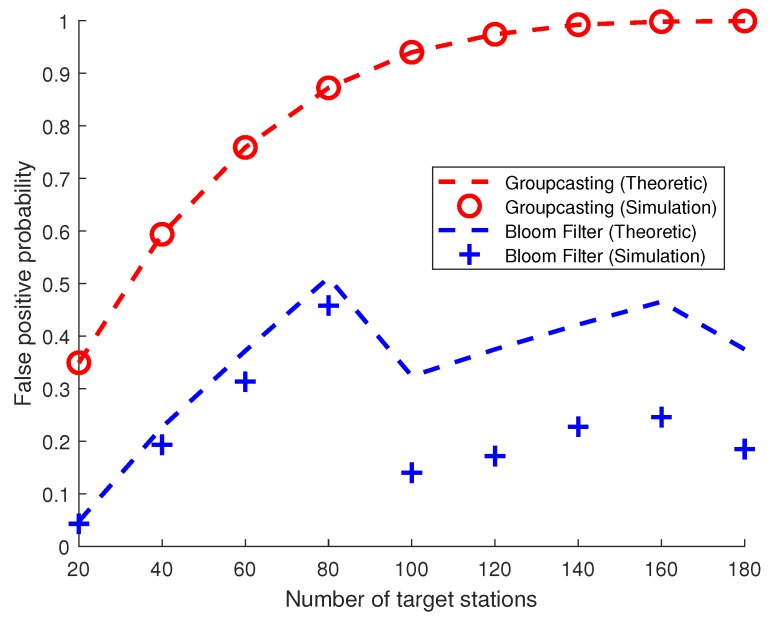
False positive probability given that the station is not the station to be woken up.

**Figure 9 sensors-19-05106-f009:**
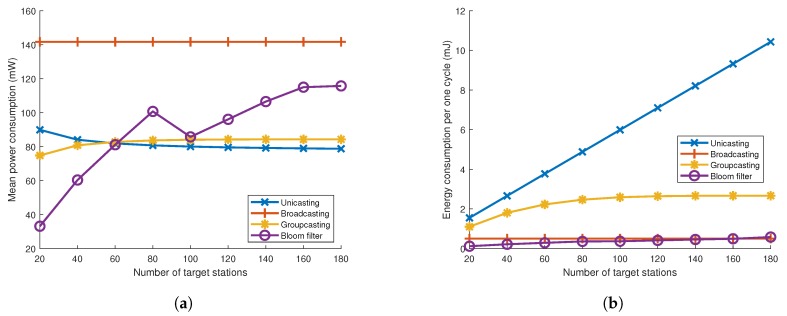
(**a**) Mean power consumption and (**b**) energy consumption until all targeted stations are waken up for each addressing mechanism.

**Figure 10 sensors-19-05106-f010:**
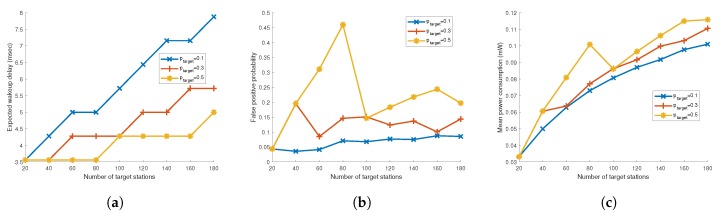
(**a**) Expected wake-up delay, (**b**) false positive probability and (**c**) mean power consumption for $p_{target}$ = 0.1, 0.3, and 0.5 for Bloom filter.

**Table 1 sensors-19-05106-t001:** Simulation parameters.

Parameter	Meaning	Value
$P_{tx,wur}$	Transmit power of WUR	88 mW
$P_{rx,wur}$	Receive power of WUR	77 mW
$P_{idle,wur}$	Idle power of conventional 802.11 radio	100 μW
$P_{tx,pcr}$	Transmit power of conventional 802.11 radio	352 mW
$P_{rx,pcr}$	Receive power of conventional 802.11 radio	154 mW
$T_{wakeupframe}$	Transmit duration for wake-up frame	704 μs
$T_{sifs}$	Short inter-frame space	16 μs
$T_{dataframe}$	Transmit duration for data frame	2.8 ms
$T_{ackframe}$	Transmit duration for ack frame	29.8 μs
$N_{station}$	Number of all stations	200
$N_{group}$	Number of groups	40
$L_{group}$	Size of a group	5
$p_{target}$	Target false positive probability	0.3

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
