# Peer review of "Performance Analysis of Addressing Mechanisms in Inter-Operable IoT Device with Low-Power Wake-Up Radio"

_sensors, 2019, doi:10.3390/s19235106_

Round 1

Reviewer 1 Report

The paper describes IEEE approach performance for wake-up technology-equipped networks in terms of latency / power / false probability wake-up / energy for networks of "n" nodes. The paper is perfectly structured and no English errors are detected.

The paper novelty is quite limited, it does not research a single novel concept. However, its results and equations are a  useful reference for (future) network deployment.

Some comments which would improve it from this reviewer's point of view are:

It should be clarified which kind of devices can benefit from this wake-up radio approach. IoT is quite a wide term. The reader could think this approach is the panacea for wake-up equipped networks but the truth is very low-power devices simply cannot use this IEEE approach because of their 1) hardware limitations 2) battery limitations. For example, cell coin powered devices simply cannot do the few hundred mW required for this approach, even for few milliseconds. The paper could benefit from a short explanation about possible architectures of the WuR radio. While the main radio is expected to be an IEEE 802.11 transceiver, it is not clear the differences in terms of hardware and or capabilities of the low-power transceiver managing wake-up calls. Figure 2 is not clear. It could benefit of some explanation or better representation of the four white stripes in the upper row. It is really not clear at which time they start appearing, and why. The paper should add a small paragraph about how mature is this technology, when is it expected to be featured by new devices, as well as some less generic possible applications of such technology.

Author Response

Response to Reviewer 1 Comments

Dear Editor and Reviewers,        

We appreciate your time taken to review our submission and thank you for the valuable comments. We have carefully revised our paper taking into consideration your comments and suggestions. The following are detailed responses to the comments. Modifications in response to the comments are marked with “bold” font in the revised manuscript.

Point 1: It should be clarified which kind of devices can benefit from this wake-up radio approach. IoT is quite a wide term. The reader could think this approach is the panacea for wake-up equipped networks but the truth is very low-power devices simply cannot use this IEEE approach because of their 1) hardware limitations 2) battery limitations. For example, cell coin powered devices simply cannot do the few hundred mW required for this approach, even for few milliseconds.

Response 1: Thank you for your valuable comments. As you mentioned, we agree that adopting IEEE 802.11ba would require redesign of hardware architecture. For example, already established IoT platform may not profit from IEEE 802.11ba design unless the legacy hardware is replaced by a new hardware which has a room for supporting additional radio. On the other hand, for example, cell coin powered devices, the lifetime can be prolonged by means of this IEEE 802.11ba approach once its hardware can support IEEE 802.11ba architecture.

We have clarified which kind of devices can benefit from this wake-up radio approach in the revision. To this end, the following paragraph has inserted.

On line 51 in Section 1: Some aspects may be required to support the WUR. Since IEEE 802.11ba belongs to 11 family, primary connected radio must support legacy IEEE 802.11. Additionally, the hardware needs to be newly deployed in order to be capable of the additional radio. However, since they are the world's most widely used wireless computer networking standards, the main framework of the IoT platform does not need to completely be redesigned. Although it also requires an additional radio, overhearing and idle listening problems can be drastically alleviated. Due to the advantages compared to duty-cycle operation, this paper considers WUR as a key solution for solving the power consumption problem of IoT devices and focuses on the detailed performance of WUR.

Point 2: The paper could benefit from a short explanation about possible architectures of the WuR radio. While the main radio is expected to be an IEEE 802.11 transceiver, it is not clear the differences in terms of hardware and or capabilities of the low-power transceiver managing wake-up calls.

Response 2: Thank you for your valuable comments. We have inserted a brief architecture of IEEE 802.11ba in Figure 3 and a short corresponding paragraph as follows:

On line 120 in Section 2.1: Figure 3 shows a brief architecture of IEEE 802.11ba. On the transmitter side, conventional 802.11 radio can transmit and receive the legacy IEEE 802.11 frames. Since the WUR does not support transmission, the transmitter does not need to be capable of reception of IEEE 802.11ba frames. On the receiver side, an additional radio is attached to the primary radio. The role of WUR is only to decode the wake-up frame and to wake up the primary transceiver. This transceiver for the receiver can be any kind of existing IEEE 802.11 standard device.

Point 3: Figure 2 is not clear. It could benefit of some explanation or better representation of the four white stripes in the upper row. It is really not clear at which time they start appearing, and why.

Response 3: Thank you for your valuable comments. The four white stripes stand for the channel access procedure, which can be distributed coordination function (DCF) or point coordination function (HCF) in IEEE 802.11. However, since the channel contention procedure is independently operated with this wake-up procedure, we have erased this part in the figure and have clarified channel access procedure as follows.

On line 115 in Section 2.1: This channel access mechanism is a part of conventional procedure of IEEE 802.11 and it can be either contention based or contention free.

Point 4: The paper should add a small paragraph about how mature is this technology, when is it expected to be featured by new devices, as well as some less generic possible applications of such technology.

Response 4: Thank you for your valuable comments. We expatiated on detailed application of IEEE 802.11ba as follows:

On line 45 in Section 1: A representative standard to support wake-up radio is IEEE 802.11ba. A task group of IEEE 802.11 was formed to develop draft 0.1 (D0.1) in 2017, and D4.0 has been released in 2019. Although the normative works of IEEE 802.11ba are just in the beginning stage, it can be applied to the scenarios of smart home/office, warehouse, outdoor cattle farms, synchronous wake-up of sensors, synchronous wake-up of wearable devices, reconnection of wearable devices, wake-up vehicle-to-pedestrian (V2P) radios, and smart scanning.

Reviewer 2 Report

The paper is very well structured and technically sound.

There have been some publications in the same domain, which the authors may include them in the reference and comment them in the introduction, e.g., https://www.semanticscholar.org/paper/A-Bloom-Filter-based-Wake-up-Communication-System-Ishida-Takiguchi/551e082c4369a61178b876981f75a93947ff0940 https://www.semanticscholar.org/paper/A-Novel-Wireless-Wake-Up-Mechanism-for-Ubiquitous-Takiguchi-Saruwatari/c11f53acb28e57608226140c12184dfe6151beaa There are several gramma errors in the manuscript.

The authors should proofreading the paper carefully. Below are just three examples:  "one of the solution" should be "one of the solutions" in page 2, line 82. "the AP first transmits wake-up frame" should be "the AP first transmits a wake-up frame". "it become" in page 16, line 465, should be "it becomes".

Since 802.11ba standard is still under development. The authors should use the information in the latest draft, currently Draft 4.0, for the paper and try to use the terminologies consistent to the spec draft. For example, The spec draft does not have "LPWUR".    The term "PCR" is no long used in the spec. And, it is never used for AP Transmitter (as shown in Figure 2)  Doze state and awake state are introduced before 802.11ba. Those terms should be carefully used.

Author Response

Response to Reviewer 2 Comments

Dear Editor and Reviewers,        

We appreciate your time taken to review our submission and thank you for the valuable comments. We have carefully revised our paper taking into consideration your comments and suggestions. The following are detailed responses to the comments. Modifications in response to the comments are marked with “bold” font in the revised manuscript.

Point 1: There have been some publications in the same domain, which the authors may include them in the reference and comment them in the introduction, e.g., https://www.semanticscholar.org/paper/A-Bloom-Filter-based-Wake-up-Communication-System-Ishida-Takiguchi/551e082c4369a61178b876981f75a93947ff0940 https://www.semanticscholar.org/paper/A-Novel-Wireless-Wake-Up-Mechanism-for-Ubiquitous-Takiguchi-Saruwatari/c11f53acb28e57608226140c12184dfe6151beaa

Response 1: Thank you for your valuable comments. We have checked your mentioned work and have cited your mentioned paper in the revision as [14]. Since the paper entitled “A Bloom Filter based Wake-up Communication System” has been cited in our initial paper, we could be able to include the paper.

[12] Takiguchi,  T.;  Saruwatari,  S.;  Morito,  T.;  Ishida,  S.;  Minami,  M.;  Morikawa,  H.    A  Novel  Wireless Wake-up Mechanism for Energy-efficient Ubiquitous Networks.  In Proceedings of 2009 IEEE International Conference on Communications Workshops (ICC), 2009, pp. 1–5.

[14] Ishida,  S.;  Takiguchi,  T.;  Saruwatari,  S.;  Minami,  M.;  Morikawa,  H.   A Bloom Filter based Wake-up Communication System.  Abstracts of IEICE TRANSACTIONS on Communications (Japanese Edition), 2011, Vol. J94-B, pp. 1397–1407.

Point 2: There are several gramma errors in the manuscript. The authors should proofreading the paper carefully. Below are just three examples:  "one of the solution" should be "one of the solutions" in page 2, line 82. "the AP first transmits wake-up frame" should be "the AP first transmits a wake-up frame". "it become" in page 16, line 465, should be "it becomes".

Response 2: Thank you for your valuable comments. We thoroughly inspected so that there was no typo or grammer error as follows. Corrected words or sentences are marked with bold.

Line 63: one of the solution -> one of the solutions

Line 83: the authors proposes -> the author proposed

Line 108: a station just go -> a station just goes

Line 112: the AP first transmits wake-up frame -> the AP first transmits a wake-up frame

Line 146: Once the receiver get -> Once the receiver gets

Line 165: the recipient also think -> the recipient also thinks

Line 174: If the hash functions has -> If the hash functions have

Line 333: the wake-up frame have -> the wake-up frame has

Line 341: We evaluates -> We evaluate

Line 394: effect -> affect

Line 441: one of the good criterion -> one of the good criteria

Line 451: does not increases -> does not increase

Line 466: it become -> it becomes

Point 3: Since 802.11ba standard is still under development. The authors should use the information in the latest draft, currently Draft 4.0, for the paper and try to use the terminologies consistent to the spec draft. For example, The spec draft does not have "LPWUR".    The term "PCR" is no long used in the spec. And, it is never used for AP Transmitter (as shown in Figure 2)  Doze state and awake state are introduced before 802.11ba. Those terms should be carefully used.

Response 3: Thank you for your valuable comments. We applied the newest spec of IEEE 802.11ba. Specifically, we use the normative languages used in the spec in the revision. Specifically, we replaced LPWUR with WUR, and PCR with conventional 802.11 radio.

We also distinguished between conventional power saving and WUR mode. Since the languages of “doze/awake state” has been used in 802.11, we clarified their power saving mode and states according to the radios. Specifically, we revised Figure 2 and corresponding paragraph as follows.

Line 113: After that, the conventional 802.11 radio of the station may let the AP know that it is ready to receive data by sending response frame (e.g., PS-Poll frame or Trigger-based frame) to the AP and this action can be a procedure of conventional power saving mode
